# Triggerable tough hydrogels for gastric resident dosage forms

Jinyao Liu[1], Yan Pang[1], Shiyi Zhang[1], Cody Cleveland [1,2], Xiaolei Yin[1], Lucas Booth[1], Jiaqi Lin[1], Young-Ah Lucy Lee[1], Hormoz Mazdiyasni[1], Sarah Saxton[1], Ameya R. Kirtane [1], Thomas von Erlach [1], Jaimie Rogner[1], Robert Langer[1,3] & Giovanni Traverso [1,2]

Systems capable of residing for prolonged periods of time in the gastric cavity have transformed our ability to diagnose and treat patients. Gastric resident systems for drug delivery, ideally need to be: ingestible, be able to change shape or swell to ensure prolonged gastric residence, have the mechanical integrity to withstand the forces associated with gastrointestinal motility, be triggerable to address any side effects, and be drug loadable and release drug over a prolonged period of time. Materials that have been primarily utilized for these applications have been largely restricted to thermoplastics and thermosets. Here we describe a novel set of materials, triggerable tough hydrogels, meeting all these requirement, supported by evaluation in a large animal model and ultimately demonstrate the potential of triggerable tough hydrogels to serve as prolonged gastric resident drug depots. Triggerable tough hydrogels may be applied in myriad of applications, including bariatric interventions, drug delivery, and tissue engineering.

[1] Department of Chemical Engineering and Koch Institute for Integrative Cancer Research, Massachusetts Institute of Technology, Cambridge, Massachusetts 02139, USA. [2] Division of Gastroenterology, Brigham and Women's Hospital, Harvard Medical School, Boston, Massachusetts 02115, USA. [3] Harvard—MIT Division of Health Sciences and Technology, Massachusetts Institute of Technology, Cambridge, Massachusetts 02139, USA. Correspondence and requests for materials should be addressed to R.L. (email: rlanger@mit.edu) or to G.T. (email: gi_lab@mailworks.org)

Devices with the ability to reside in the gastric cavity have been applied for diagnosis[1–3] and treatment interventions in humans[4–7]. The materials used for their fabrication have been largely limited to thermoplastic and thermosets[8–10]. Expanding the repertoire of materials available for system fabrication stands to expand the potential set of applications by enabling greater compatibility with gastrointestinal (GI) tissues, and, moreover, potentially imparting significantly improved safety. When considering future development of novel orally administered systems for prolonged drug release, a key set of elements needs to be considered to ensure safety and potential efficacy of such systems, which are destined to reside in the gastric cavity[11–13]. These include dosage forms in a form factor compatible with ingestion and subsequent shape change to enable gastric residence and prevent passage through the pylorus (the combination of these two elements is analogous to the "ship in the bottle" problem), triggerable elements, which can induce controlled breakage of the dosage form, which can be either administered or are induced by virtue of anatomic location, mechanical, and material properties to ensure stability and integrity in the GI environment, and the ability to be loaded with drug and release drug in a controlled sustained manner.

One potential set of materials with enhanced biocompatibility are hydrogels because of their softness, which can potentially minimize mucosal damage, low polymer content that may reduce the side effects of materials as well as maximize the capacity for dehydration and rehydration in form factors compatible with ingestion and subsequent gastric retention upon swelling, and release drugs in controllable manners[14–17]. Conventional hydrogels though generally suffer from being weak and, therefore,

can be easily broken by the significant compressive and shearing forces of the GI tract limiting their stability in this environment[18]. This is exemplified by the application of high viscosity grade cellulose polymers, polyacrylic acid, polyacrylates, polyacrylamides, and polyethers, which have been applied to extending release of drugs on the order of 12–16 h[19]. Though these systems can provide extension of drug release, they have only enabled the extension of release by hours[20]. A set of materials with the mechanical properties capable of potentially surviving the gastric environment are tough hydrogels. Though significant advancement has been made in fabricating hydrogels with tougher characteristics, including double-network[21–25], topological[26–30], and nanocomposite hydrogels[31–34], these gels lack the capacity to be triggered to dissolve in physiological environments, an important property to ensure the maximal safety and removal of such a system that could reside in the gastric cavity for multiple days. Long-term gastric resident systems can be associated with gastric outlet obstruction and/or the development of an allergic reaction could ensue from a gastric drug depot. These complications would require immediate clinical intervention and therefore a system that can be triggered to dissolve and thereby enable passage through the GI tract could have significant advantages over ones without the capacity for triggering where a procedural intervention-like surgery or endoscopy would otherwise be required for removal. Therefore, one essential aspect of safety lies in the ability to trigger the dissolution and controlled fracture of the material to enable rapid transit through the GI tract.

Here we describe a novel set of materials, triggerable tough hydrogels (TTHs), containing up to ~90% water enabling

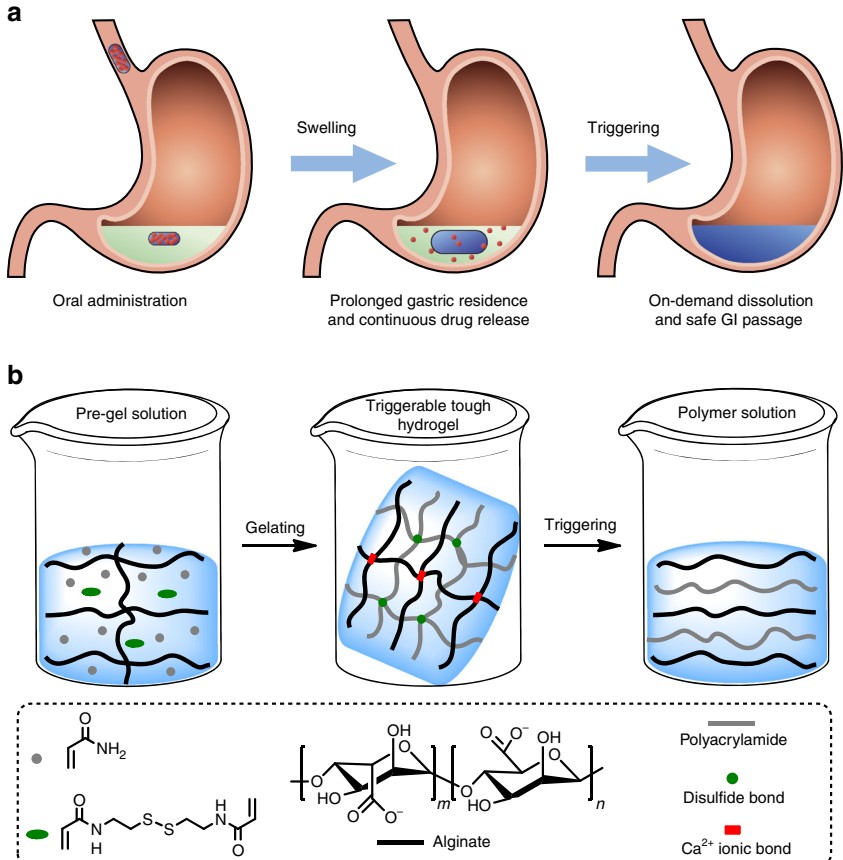

**Fig. 1** TTH platform and synthesis approach. **a** Gastric resident dosage forms for prolonged drug delivery. **b** TTH dosage form concept and synthesis design. TTHs consist of two types of alginate and polyacrylamide networks that are intertwined, and separately crosslinked by stimuli-responsive Ca²⁺ ionic and disulfide bonds, which can be dissolved into solution with a biocompatible chelator and reducing agent

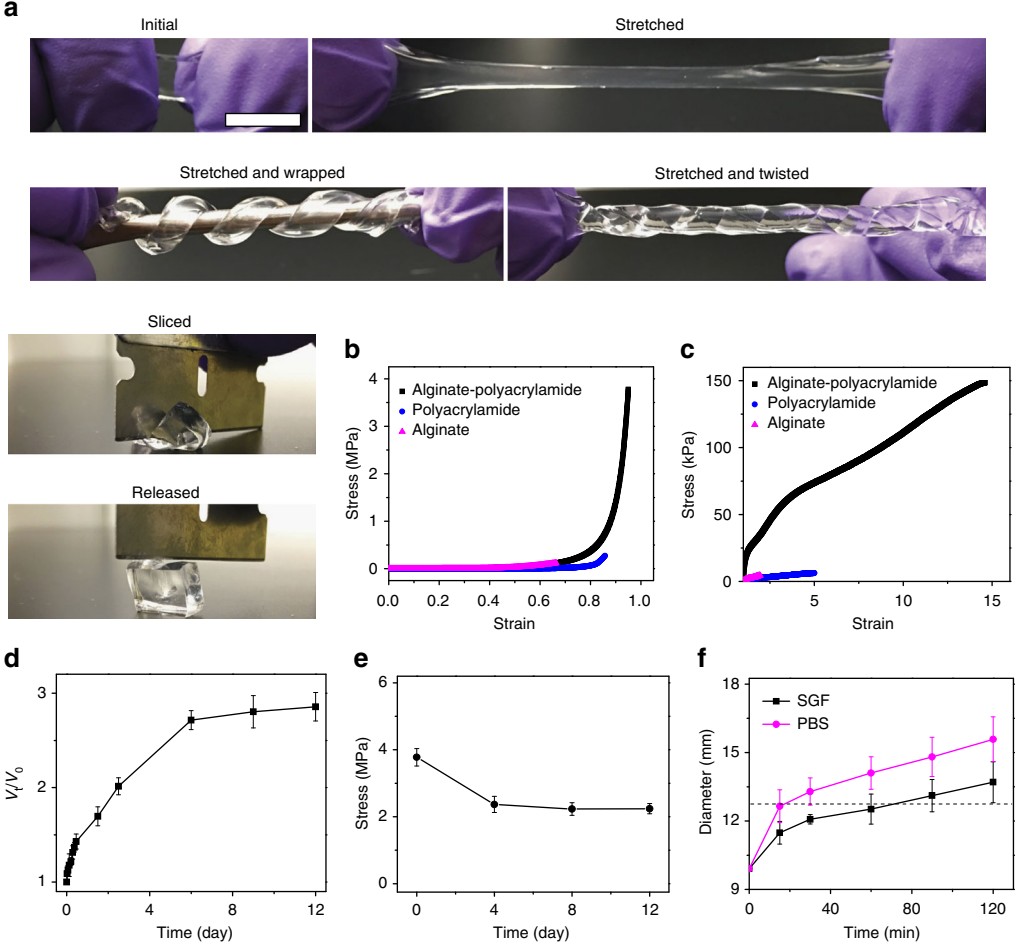

**Fig. 2** Physical characterization of TTHs. **a** Photographs of a TTH strip stretched to 14 times its initial length and subsequently coiled and twisted, and a TTH cuboid-resisted slicing with a blade. *Scale bar*: 1 cm. **b** Compressive stress-strain curves of the TTH, alginate, and polyacrylamide gels with same amounts of alginate or polyacrylamide to the TTH. **c** Tensile stress-strain curves of the TTH, alginate, and polyacrylamide gels stretched to breaking. **d** Plot of volume variation ($V_t/V_0$) of the TTH vs. the incubation time at 37 °C. **e** Plot of the maximum compressive stress of the TTH as a function of the incubation time in SGF at 37 °C. **f** Plot of diameter variation of the cylindrical dehydrated TTH vs. the incubation time at 37 °C. *Error bars* show standard deviation ($n = 3$)

significant shrinkage and swelling to address the "ship in the bottle" challenge. These gels demonstrate stability in normal gastric environments and furthermore can be triggered to dissolve rapidly with biocompatible agents, enabling the development of gastric devices with precisely controlled residence times. We further demonstrate a proof-of-concept gastric resident drug delivery system from these materials in a large animal model. Specifically, the TTHs consist of two types of polymeric networks that are intertwined and separately crosslinked by stimuli-responsive bonds, which can be triggered to dissolve into solution with biocompatible agents on demand. Given the tough and triggerable properties of these hydrogels, we anticipate their application in a broad set of GI biomedical applications.

## Results

**Preparation and characterization of TTHs.** TTHs consisting of alginate and polyacrylamide networks are crosslinked by ionic $Ca^{2+}$ and disulfide bonds, respectively (Fig. 1). Alginate is a linear copolymer comprised of blocks of α-L-guluronic acid, β-D-mannuronic acid, or alternating α-L-guluronic and β-D-mannuronic acids. Divalent $Ca^{2+}$ cations can crosslink alginate by simultaneously associating with carboxylic groups in the α-L-guluronic acid blocks from different alginate chains, forming an

ionically crosslinked network in water[21]. By contrast, the polyacrylamide network can be formed by aqueous radical polymerization of acrylamide using a bifunctional monomer as the crosslinker. Since alginate and polyacrylamide networks are separately crosslinked, stimuli-responsive ionic and disulfide bonds can be incorporated making the gels susceptible to degradation by biocompatible chelators and reducing agents, TTHs can be de-crosslinked and dissolved into solution accordingly. Alginate and polyacrylamide were selected given their well-recognized biocompatibility and broad biomedical application[35].

TTHs were fabricated by a simple one-step method. We dissolved all ingredients needed to form the two networks in deionized water, including sodium alginate and an ionic cross-linker of calcium sulfate for the ionically crosslinked alginate, as well as acrylamide, crosslinking monomer N,N′-bis(acryloyl) cystamine, thermo-initiator of ammonium persulfate, and polymerization accelerator of N,N,N′,N′-tetramethylethylenediamine for the disulfide crosslinked polyacrylamide. The mixture was heated to 50 °C for 1 h and then left in a humid box for 1 day. The unreacted ingredients were purified by continuous extraction with water demonstrating elimination of the acrylamide monomer (Supplementary Fig. 1). Details of the synthesis and characterization of TTHs are described in the Supplementary Information.

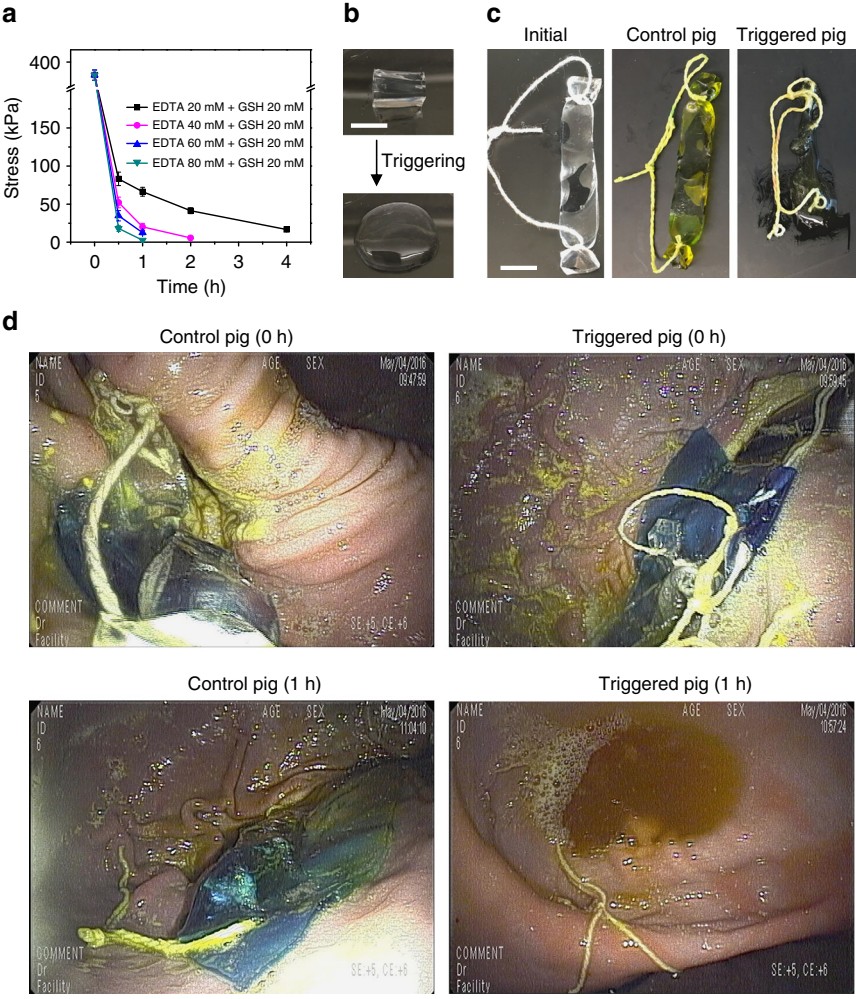

**Fig. 3** Triggerable properties of TTHs. **a** Plot of compressive stress of the TTH at strain of 80% vs. the incubation time with EDTA and GSH at 37 °C. *Error bars* show standard deviation ($n = 3$). **b** Pictures of the TTH dissolved into viscous solution after 1 h incubation with 80 mM of EDTA and 20 mM of GSH. **c** Photographs of the initial TTH strip before administration, and the retrieved TTH strips after 1 h residence in the gastric cavity of the control and triggered pigs, respectively. **d** Endoscopy images of the TTH sheets in the stomach from the control and triggered pigs, respectively. The TTH sheets were labeled with methyl blue. The pigs were treated with 40 mM of EDTA and 20 mM of GSH after delivery of the TTH strips or sheets through the esophagus. Control animals did not receive EDTA/GSH. *Scale bar*: 1 cm

The TTH synthesized had a water content of 87%, was highly stretchable, flexible, and could not be easily cut with a blade (Fig. 2a). It achieved a maximum compressive stress of $3.78 \pm 0.26$ MPa that was 14 and 32 times higher than the hydrogels composed of polyacrylamide ($0.275 \pm 0.033$ MPa) or alginate ($0.121 \pm 0.017$ MPa) alone (Fig. 2b). The tensile strength and fracture strain were, respectively, $149 \pm 11$ kPa and $14.6 \pm 1.3$ for the TTH, $6.2 \pm 0.8$ kPa and $5.0 \pm 0.6$ for the polyacrylamide gel, and $4.2 \pm 0.7$ kPa and $1.9 \pm 0.3$ for the alginate gel (Fig. 2c). The energy dissipation of the TTH was further tested by loading-unloading experiments, showing that the TTH dissipated energy effectively, as verified by the notable hysteresis, while the permanent deformation after unloading was negligible, as demonstrated by loading several samples to large values of stretch before unloading (Supplementary Fig. 2).

We next studied the swelling behavior and variation of mechanical properties of TTHs in simulated gastric fluid (SGF, pH = ~1.2). The TTH swelled progressively (Fig. 2d) and a plateau of volume variation ($V_t/V_0$) of $2.7 \pm 0.15$ was reached after 6 days of incubation at 37 °C with an accompanying decrease of the tensile properties at rupture. The tensile strength and fracture strain of the TTH decreased to $74.1 \pm 6.7$ kPa and

$12.2 \pm 1.1$, $62.4 \pm 4.9$ kPa and $10.1 \pm 0.8$, as well as $49.2 \pm 5.7$ kPa and $8.8 \pm 0.9$ after incubated for 4, 8, and 12 days, respectively (Supplementary Fig. 3a). The swelling also adversely affected the maximum compressive stress of the TTH, which decreased to $2.24 \pm 0.24$ MPa after 4 days of incubation in SGF (Fig. 2e). After the initial decrease mainly attributed to swelling; however, the maximum compressive stress of the TTH appeared to plateau with further incubation, as confirmed by a small change from $2.21 \pm 0.18$ to $2.07 \pm 0.15$ MPa between 8 and 12 days of incubation, respectively. It is worth mentioning that the maximum gastric pressure in the fasted and fed states in humans is known to range from 0.01 to 0.013 MPa[36, 37], which is far lower than the maximum compressive stress of the TTH even following incubation periods of up to 12 days, suggesting the potential of these gels to resist gastric compression and achieve relative long-term residence in the gastric cavity. We further measured the dehydration and rehydration of TTHs and found that air-drying effectively dehydrated and shrunk the gel significantly (Supplementary Fig. 3b). Scanning electron microscopy (SEM) images displayed a uniform structure of the dried TTH sample (Supplementary Fig. 4a). As expected, the TTH could not dehydrate into a smaller size by lyophilization and a

micro-porous structure was obtained for the lyophilized sample (Supplementary Fig. 4b, c). We next measured the rehydration of TTHs in SGF and found that a completely dehydrated TTH with similar dimensions to a standard 000 capsule swelled to a size greater than the diameter of the resting human pylorus $(12.8 \pm 7.0 \text{ mm})$[38, 39] within 70 min (Fig. 2f), which is within the 50th percentile for gastric emptying in humans[40]. In addition, the dehydrated TTH could swell to a size larger than the diameter of pylorus within 15 min in a neutral pH approximating the fed state or patients taking antacids or that can be achieved by co-administration with antacids. The enhanced swelling is attributed to the higher solubility of alginate in neutral pH than in an acidic environment. We found that the adequately rehydrated gel demonstrated a maximum compressive stress of $2.02 \pm 0.18$ MPa (Supplementary Fig. 5), demonstrating the retention of toughness of TTHs after a cycle of complete dehydration and subsequent rehydration. Alternatively, a TTH-based encapsulation system encasing $CaCO_3$ in an initial form factor of a standard 000 capsule swelled to 27 mm within 30 min in SGF (Supplementary Fig. 6a, b). Similar strategies can be applied for enabling flotation of TTHs (Supplementary Fig. 6c). Given the highly stretchable and tough characteristics, various dosage forms compatible with ingestion and subsequent gastric residence through size exclusion could be developed by using TTHs.

Initial biocompatibility of TTHs was evaluated through in vitro cell toxicity analysis. The gel was incubated in cell culture medium across a wide range of concentrations from 0.2 to 50 mg ml$^{-1}$ at 37 °C for 24 h. The medium was then tested for its cytotoxicity on multiple cell lines, including HeLa, Caco-2 (C2BBe1 clone), and HT29-MTX-E12 (Supplementary Fig. 7). No significant cytotoxicity was observed for the medium incubated with the gel in any of these cell lines at the end of a 24 h culture period. We next conducted extended cytotoxicity analysis by culturing the TTHs with intestinal stem cells (ISCs) and demonstrate excellent cytocompatibility of the TTHs with mouse Lgr5$^+$ stem cells over the course of 5 days (Supplementary Fig. 8a). Furthermore, we showed that Lgr5$^+$ stem cells could be cultured on and within TTHs and these retained their ability of multilineage differentiation to form organoids (Supplementary Fig. 8b), supporting the biocompatibility and potential application of TTHs serving as a substrate for organoid culture.

**Triggerable properties of TTHs**. We next investigated the stimuli-responsiveness of TTHs by using ethylenediaminetetraacetic acid (EDTA) and glutathione (GSH) as triggers of the $Ca^{2+}$ ion and disulfide crosslinks. Both EDTA and GSH have been previously used in humans as treatments or supplements with oral dosages of up to 6 and 5 g daily, respectively[41, 42]. In addition, we recognized that both EDTA and GSH have been used as additives in foods[43, 44]. To measure the triggerable properties and potential boundaries set by EDTA and GSH found in a human diet, TTHs were incubated at 37 °C with a range of concentrations from 20 to 80 mM of EDTA and GSH well above the concentrations found in food for various time intervals, and then evaluated for compressive stress to characterize the dissolution behavior of the gels. Interestingly and supporting the selectivity of the EDTA and GSH combined triggering solution, the TTH could not be dissolved by incubation with EDTA or GSH alone even when incubation times were increased to 24 h (Supplementary Fig. 9). These data support the ability to maintain a network by the remaining crosslinked single network hydrogel and demonstrate that de-crosslinking of both alginate and polyacrylamide networks are essential to dissolve the TTH. The requirement for both EDTA and GSH for triggering supports

the likely sustained stability of the TTH in the presence of a normal human diet. The dissolution of the gels was accelerated by triggering with EDTA and GSH simultaneously. As shown in Fig. 3a, the compressive stress of the TTH decreased rapidly from $373 \pm 10$ to $66.3 \pm 5.8$, $41.3 \pm 3.9$, and $16.7 \pm 1.0$ kPa after 1, 2, and 4 h incubation in 20 mM of EDTA and 20 mM of GSH. When the concentration of EDTA was increased to 40 mM while the GSH was kept constant, the compressive stress of the TTH reduced dramatically to $5.6 \pm 0.04$ kPa and the gel dissolved after 2 h of incubation. The TTH started to dissolve into a viscous solution only after 1 h incubation with further increases of EDTA to 80 mM (Fig. 3b). In contrast, increases in GSH concentration retarded the dissolution of the TTH (Supplementary Fig. 10), suggesting that the carboxyl group at the C terminus of GSH could disturb the formation of the ionic bond between the $Ca^{2+}$ and the carboxyl groups in EDTA when excessive GSH was present. Gel permeation chromatography (GPC) of the dissolved TTH demonstrated two peaks with molecular mass of ~120 and ~200 kDa that corresponded to the dissociated alginate and polyacrylamide chains, respectively, supporting the dissolution of the TTH into free polymers (Supplementary Fig. 11). In vitro cell viability assays verified the low cytotoxicity of these dissociated free polymers against HeLa, Caco-2, and HT29 cell lines at the end of a 24 h culture with concentrations up to 5 mg ml$^{-1}$ (Supplementary Fig. 12).

Having confirmed in vitro the superior stimuli-responsiveness of the gels, we next tested the in vivo dissolution of TTHs by using a Yorkshire pig animal model, which has been previously established for the evaluation of GI resident systems[13, 45]. Yorkshire pigs weighing 45–55 kg have gastric and intestinal anatomy and dimensions similar to humans[46]. TTH strips with dimensions 50 mm × 10 mm × 5 mm were introduced endoscopically into the stomach. Pigs were administered a triggering solution consisting of 0.5 L of EDTA (40 mM) and GSH (20 mM) after deployment of the TTH strips. Control samples were deployed into the stomach without the addition of the triggering solution. The TTH strips were retrieved endoscopically after 1 h in the gastric cavity. Strips retrieved from the control pigs remained intact and retained a maximum compressive stress of $1.77 \pm 0.15$ MPa (Supplementary Fig. 13), whereas the strips from the treated pigs dissolved into viscous solution (Fig. 3c). To further view the in situ dissolution of TTHs in stomach, large TTH sheets, in the shape of an equilateral triangle (side length, 10 mm; thickness, 3 mm) were prepared and labeled with methyl blue. These were triggered in situ with the EDTA/GSH solution and endoscopic videography was used for image capture. Endoscopic video revealed that the TTH sheets were triggered to dissolve within 1 h in the gastric cavities of the treated pigs, whereas the sheets in the control pigs remained intact (Fig. 3d). These results support that TTHs can be triggered to dissolve in vivo with biocompatible agents. We note that long-term (>24 h) gastric resident systems present risks to patients including GI mechanical obstruction and the inability to discontinue a drug in the event of developing an allergic reaction through non-invasive means. The ability to trigger the dissolution of such systems is therefore essential for safe clinical implementation. The need for triggering is further amplified in resource-constrained settings, where healthcare interventions-like endoscopy and surgery may be largely limited and where the inability to remove such systems could manifest in significant morbidity and mortality.

**Gastric retentive drug delivery of TTHs**. To evaluate the mechanical integrity of TTHs and their potential application as triggerable biomedical materials in gastric resident systems, we

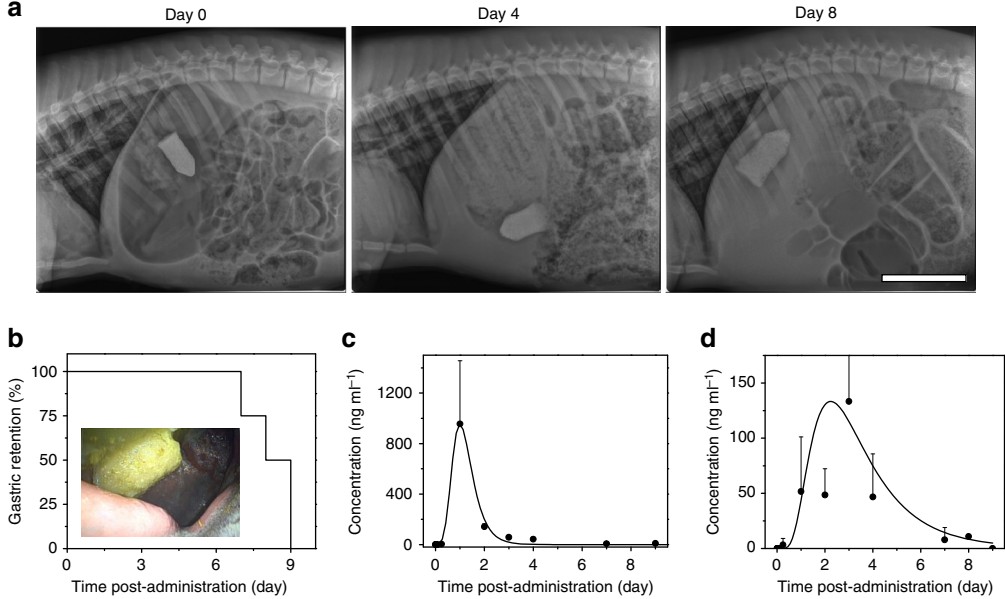

**Fig. 4** Gastric retention and drug release of TTHs in pigs. **a** Representative X-ray images of a TTH device residing in the gastric cavity of a Yorkshire pig. *Scale bar*: 10 cm. **b** Plot of remaining percentage of the intact TTH devices in the pig stomach monitored by X-ray imaging vs. time post administration (the *inset* represents endoscopic image of the TTH device after 8 days retention in the gastric cavity). Plot of blood-drug concentration as a function of time post administration: **c** free lumefantrine; **d** the lumefantrine-loaded TTH device. In the pig experiments, one TTH device per pig was implanted at day 0 through the esophagus. The capsule-like TTH devices with an initial volume of 22 ml (diameter, 2.8 cm; total length, 5 cm) were labeled by radiopaque barium sulfate for the gastric residence evaluation or loaded with lumefantrine for the pharmacokinetic study. *Error bars* show standard deviation ($n = 3$-4)

fabricated TTH prototypic gastric resident dosage forms. These were evaluated for their gastric residence and integrity in Yorkshire pigs. To evaluate the gastric retention and in vivo integrity of TTHs, radiopaque capsule-like TTH dosage forms with volumes of 22 ml (diameter, 2.8 cm; total length, 5 cm) were designed and prepared by mixing barium sulfate with the pre-gel solution immediately prior to polymerization. It was noted that the significant load of barium sulfate required for radiographic visualization (20 wt%) manifested in slower swelling characteristics than the TTHs (Supplementary Fig. 14). We therefore used barium sulfate-containing TTHs in their hydrated states, which enabled the retention by virtue of the size of the gel administered and radiographic visualization by virtue of their barium content. Four individual experiments in four different pigs were performed and radiographs were taken approximately every 48–72 h to monitor the integrity of the dosage form, its anatomic location, and any evidence of GI obstruction. Intact prototype TTH systems were observed to achieve gastric residence of 7-9 days (Fig. 4a, b). TTHs remained stable in vitro in SGF (>12 days) though in vivo breakage of TTHs was observed earlier than this likely due to the compressive stress associated with gastric motility, and potential de-crosslinking of alginate network by exchange reactions with monovalent cations in the GI environment[47]. Meanwhile, the disulfide bonds in polyacrylamide network may be reduced by protein or peptide-associated thiols though low-molecular weight thiols, GSH, and cysteine are only present at a low level or even absent in human gastric fluid[48]. No intact devices were visualized outside of the gastric cavity, supporting that device breakage first occurred in the stomach enabling their eventual passage out of the stomach. Once device breakage occurred, the resulting fragments were visualized in the intestines without evidence of intestinal obstruction (Supplementary Fig. 15). Throughout the experiments, the animals were found to have normal eating and stooling patterns and did not exhibit any signs of GI obstruction, either clinically or radiographically.

Medication non-adherence is a major challenge for the treatment of malaria and having the capacity to deliver drugs in a single administration event has the potential to not only enhance cure rates in acute malaria, but also decrease resistance rates. To demonstrate a potential application of this system, a gastric resident TTH dosage form containing lumefantrine, a hydrophobic antimalarial drug, was selected to study the drug loading and release from the TTH material. The lumefantrine-loaded TTHs were similarly fabricated by mixing drug powder with the prepolymerization solution just before gelation. The degree of drug loading was easily controlled by adjusting the feed ratio of drug. The maximum compressive stress of the gel increased from $3.91 \pm 0.31$ to $5.43 \pm 0.61$ MPa with the increase of drug loading from 1 to 10 wt%, whereas the fracture strain decreased from $14.7 \pm 1.3$ to $11.9 \pm 1.5$ and the tensile strength remained around $180 \pm 20$ kPa (Supplementary Fig. 16). In vitro release kinetics of the lumefantrine dosage forms were characterized under predetermined sink conditions[49] and the results showed that the release of lumefantrine could be controlled by tuning the drug loading. In vitro characterization of the cumulative release of lumefantrine after 12-day incubation in SGF increased from $8.3 \pm 0.17$ to $61 \pm 3.7\%$ with the decrease of drug loading from 10 to 1 wt%, suggesting the diffusion of drug was decreased as a function of reduced swelling of the TTH associated with the increase in hydrophobicity of the gels from the higher lumefantrine load (Supplementary Fig. 17a, b). A first-order rate equation was fit to describe the rate of drug release, and the release rate constants from gels loaded with 1, 5, and 10% drug were found to be 11.1, 0.2, and 0.36 day$^{-1}$, respectively. It was noted that the post-polymerization purification affected the drug loading of TTHs prepared by mixing drug powder with the prepolymerization solution. An alternative strategy was demonstrated to avoid drug loss during the preparation of drug-loaded TTHs by using post-polymerization encapsulation. As shown in Supplementary Fig. 17c, we first prepared the purified TTH, then lyophilized and subsequently

rehydrated the TTH in the aqueous solution of drugs. In addition, to evaluate the potential delivery of a range of molecules, transport of model molecules across a range of molecular weights was evaluated through TTHs. Specifically, insulin, rifampicin, and dimethyl sulfoxide (DMSO) were observed to transport efficiently through the TTH and showed size-dependent permeability that increased from 0.016, 0.042 to 0.082 ml h$^{-1}$ cm$^{-2}$ with the decrease of molecular weight from 5808, 823 to 78 Da (Supplementary Fig. 18). To evaluate the release kinetics from TTH in vivo, lumefantrine-loaded TTH systems in the same dimensions and shape to the TTH system used for the gastric retention and integrity studies were prepared. The pharmacokinetic studies were carried out by single administration of one drug-loaded TTH device containing 960 mg of lumefantrine per pig. The in vivo pharmacokinetics were significantly extended when administered in the form of TTH as compared to the unformulated free drug control (Fig. 4c, d). After a single administration of free lumefantrine, the drug was rapidly cleared from blood with a rapid terminal elimination phase (Supplementary Fig. 19). In contrast, a relative constant blood drug concentration remained up to 4 days after a single administration of the lumefantrine-loaded TTH device, supporting the potential for multi-day dosing using the TTH drug delivery system. A pharmacokinetic model described by first-order rate equations was fit to the data. The absorption rate constant for both formulations was 1.17 day$^{-1}$. The rate constant for drug release in vivo was estimated to be 0.68 day$^{-1}$, which is ~threefold higher than the in vitro release rate constant. This may be because in vitro tests do not account for food effects and other gastric secretions, which may significantly affect drug release. The elimination rate constants for the free drug was estimated to be 1.17 day$^{-1}$ and that apparent elimination rate constant of the drug delivered in TTH was 0.68 day$^{-1}$ indicating delayed elimination.

In summary, we present a novel family of TTHs and demonstrate their capacity for significant dehydration and rehydration. We further demonstrate their capacity to be triggered to dissolve with the application of biocompatible triggers. TTHs were evaluated for their stability and mechanical integrity in a large animal model. A potential application in drug delivery was also demonstrated with an extended release system for lumefantrine. Preclinical studies will be required to translate these systems for human application including further safety studies and stress testing in other large animal models. In sum, the TTHs described herein present three important points of novelty from the hydrogel perspective: exceptional mechanical properties: that can withstand in vivo gastric forces and achieve long-term residence in the stomach of a large mammal; remarkable triggerable properties: capable of on-demand dissolution; TTHs can be drug loaded and provide controlled drug release. We believe that this combination of features makes TTHs uniquely attractive for the development of advanced gastric dosage forms for prolonged drug delivery, ingestible electronics, and bariatric applications.

## Methods

**Materials**. Acrylamide (A9099, ≥99%), N,N′-bis(acryloyl)cystamine (A4929), ammonium persulfate (A3678, ≥98%), N,N,N′,N′-tetramethylethylenediamine (T9281, 99%), sodium alginate (A2033, medium viscosity), calcium sulfate (C3771, ≥99%), methyl blue (M6900), barium sulfate (11844), L-GSH reduced (≥98%), EDTA (≥99%), DMSO (D8418), sodium bicarbonate (NaHCO$_3$, S5761), calcium carbonate (CaCO$_3$, ≥99%), and phosphate-buffered saline (PBS, pH 7.4) were available commercially from Sigma-Aldrich and used as received unless otherwise noted. Insulin was kindly provided by Novo Nordisk and labeled by Alexa-Fluor® 488. Lumefantrine and rifampicin were purchased from Hangzhou Hysen Pharma Co., Ltd in China. Nanopure water (18 MΩ cm) was acquired by means of a Milli-Q water filtration system, Millipore (St. Charles). SGF (pH ~1.2) was made

by dissolving 2 g NaCl and 8.3 ml concentrated HCL in nanopure water and adjusting to 1000 ml.

**Mechanical characterization**. The mechanical characterization in tension and compression was performed on an Instron testing machine according to ASTM standards D638 (tension) and D575 (compression). For tensile measurement, specimens were loaded into the grips with a 50 N load cell and the gauge length measured using a digital micrometer. Displacement was applied to the specimen at a rate of 0.15 mm s$^{-1}$ until samples ruptured. For compression measurement, specimens were placed into a constrained loading compression jig with a 500 N load cell and the gauge length measured using a digital micrometer. Displacement was applied to the specimen at a rate of 0.05 mm s$^{-1}$ until reaching 95% compressive strain. Force was converted into pressure ($F/A$) and displacement into strain ($\Delta L/L$).

**High performance liquid chromatography (HPLC)**. HPLC measurement was carried out on an Agilent 1260 Infinity HPLC system equipped with a quaternary pump, autosampler, thermostat, control module, and diode array detector as described previously[45] with minor modifications. The output signal was monitored and processed using the ChemStation® software. Chromatographic separation was carried out on a 50 mm × 4.6 mm EC-C18 Agilent Poroshell 120 analytical column with 2.7 µm spherical particles, maintained at 40 ℃. The optimized mobile phase consisted of acetonitrile, methanol, and buffer (pH 3.5 adjusted with 0.1% formic acid) (72:20:8, v/v) at flow rate of 0.5 ml min$^{-1}$ over a 10 min run time. The injection volume was 4 µl, and the ultraviolet (UV) detection wavelength of 254 nm was selected.

**Liquid chromatography tandem-mass spectrometry**. UPLC separation was conducted on a Waters UPLC aligned with a Waters Xevo-TQ-SMS mass spectrometer (Waters Ltd., UK) as described previously[45] with minor protocol modifications as follows. MassLynx 4.1 software was used for data acquisition and analysis. Liquid chromatography separation was performed on an Acquity UPLC CSH C18 (50 × 2.1 mm, 1.7 µm particle size) at 50 ℃. The mobile phase consisted of acetonitrile, 0.1% formic acid, and 10 mM ammonium formate was flowed at a rate of 0.6 ml min$^{-1}$ using a time and solvent gradient composition. The initial gradient (100%) was followed by a linear gradient (20%) over 0.25 min. Over the next 1.25 min, the gradient was brought to 0% and held for 0.5 min and finally brought back to the initial gradient of 100% over 0.25 min and held until the end of the run for column equilibration. The total run time was 4 min and sample injection volume was 2.5 µl. The mass spectrometer was operated in the multiple reaction-monitoring mode. Sample introduction and ionization was ESI in the positive ion mode. Stock solutions of lumefantrine and an internal standard artemisinin were prepared in methanol at a concentration of 500 µg ml$^{-1}$. A ten-point calibration curve was prepared ranging from 2.5 to 2500 ng ml$^{-1}$. Quality-control samples were prepared in a similar procedure using an independent stock solution at three concentrations (2.5, 25,and 250 ng ml$^{-1}$). A measure of 200 µl of internal standard 250 ng ml$^{-1}$ was added to 100 µl of sample solution to cause precipitation. Samples were vortexed and sonicated for 10 min and then placed in a centrifuge for 10 min. A measure of 200 µl of solution was pipetted into a 96-well plate containing 200 µl of water. Finally, 2.5 µl was injected into the Ultra Performance Liquid Chromatography-electrospray ionization-MS system for analysis.

**SEM**. Surface morphology of the dehydrated gels was observed using the JEOL 5600LV SEM. For visualization under SEM, samples were fixed to aluminum stubs with double-sided adhesive carbon conductive tape and subsequently sputter coated with carbon using a Hummer 6.2 Sputter System.

**GPC**. Aqueous GPC was conducted on a Viscotek system (Malvern) equipped with an isocratic pump Viscotek VE 1122 solvent delivery system, TDA 305 triple detector array, and three TSK Gel GMPWxL column with guard column. The system was equilibrated at 30 ℃ in pre-filtered water containing 0.05 M NaNO$_3$ with the flow rate set to 1 ml min$^{-1}$. Polymer solutions were prepared at a concentration of about 0.5-5 mg ml$^{-1}$ and an injection volume of 200 µl was used. Data collection and analysis were performed with ChemStation for LC (Agilent) and OmniSEC v. 4,6,1,354 software (Malvern). The system was calibrated with poly (ethylene oxide) standards (Sigma) ranging from 400 to 511,000 Da ($M_p$).

**Preparation of TTHs**. TTHs were prepared by a one-pot synthetic method. Typically, acrylamide (3.60 g, 50.6 mmol), N,N′-bis(acryloyl)cystamine (13.2 mg, 0.051 mmol), ammonium persulfate (57.8 mg, 0.253 mmol), and sodium alginate (600 mg) were dissolved into 30 ml nanopure water. N,N,N′,N′-tetra-methylethylenediamine (29.4 mg, 0.253 mmol) and calcium sulfate (120 mg, 0.697 mmol) were added after a homogeneous solution was obtained. Calcium sulfate was added as a suspension into the reaction mixture because of its limited water solubility caused by its low dissociation constant. Although the association of Ca$^{2+}$ with the carboxyl groups in alginate could accelerate the dissolution of calcium sulfate, the complete dissolution took place overnight. Thus, the reaction mixture

was presented as a free solution before subjecting it to polymerization even after all the ingredients were added. The solution was carefully degassed and then quickly poured into standard dumbbell die (ASTM D-638) molds. The gel was crosslinked by heating to 50 °C for 1 h, then sitting in a humid box at room temperature for another 24 h to stabilize the reaction. Afterwards, the resulted TTHs were subjected to mechanical characterization. To prepare the TTH membrane for permeability measurement, the pre-gel solution was poured into a glass mold covered with a 3-mm-thick glass plate. To prepare a TTH-based floating system, CaCO₃ powder (5 wt%) was added into the reaction mixture just before polymerization. For in vivo dissolution study, the TTH membrane was labeled with methyl blue by adding a drop of dye solution onto the top of the TTH membrane and then covered by a glass plate and further incubated overnight. To prepare radiopaque-labeled capsule-like TTHs, 20 ml pre-gel solution containing barium sulfate (20 wt%) was added into a 50 ml CORNING CentriStarTM tube immediately prior to polymerization. The drug-loaded TTHs were similarly fabricated by mixing lumefantrine powder with the prepolymerization solution just before gelation, and the degree of drug loading was easily controlled from 1 to 10 wt% by adjusting the feed ratio of drug. To prepare water soluble drug-loaded TTHs, the purified TTH was lyophilized and subsequently rehydrated in the aqueous solution of rifampicin (a water-soluble antibiotic).

**Purification of TTHs**. To measure the unreacted ingredients in TTHs, the resulted gel was cut into 1–2 mm pieces and sonicated in 10 volumes of water for 30 min. The mixture was further incubated at 37 °C for 24 h on a shaker plate at 250 r.p.m. After the addition of a certain volume of acetonitrile, the mixture was centrifuged and the supernate was analyzed by HPLC. To purify the TTH, the obtained gel was extensively extracted with 4 × 1000 ml water for 24 h. The same procedure described above was performed to measure the unreacted ingredients in the purified TTH.

**Swelling and stability of TTHs in SGF**. The swelling and stability of TTHs were measured by incubating TTH samples in SGF at 37 °C and subsequent measuring the volume as well as the maximum compressive stress. Typically, the cylindrical TTH samples (diameter, 6.2 mm; length, 12 mm) were prepared by carrying out the gelation reaction in a 3.5 ml VWR glass vial. The obtained gels were submerged in 50 ml GSH in a Corning CentriStarTM tube and then incubated at 37 °C on a shaker plate at 250 r.p.m. After predetermined time intervals, the size of the samples was measured by using a digital micrometer and compared with initial volumes. Meanwhile, the TTH samples were also subjected to compression measurement. Three replicates were conducted for each TTH sample.

**Dehydration and rehydration of TTHs**. The dehydration of TTHs was measured by incubating TTH samples in air at 37 °C. Typically, the cylindrical TTH samples (diameter, 6.2 mm; length, 12 mm) were placed in the oven set at 37 °C and the size of the samples after predetermined incubation intervals was measured by using a digital micrometer and compared with initial volumes. For rehydration measurement, the dehydrated gel samples were submerged in 50 ml SGF in a Corning CentriStarTM tube and incubated at 37 °C on a shaker plate at 250 r.p.m. After different time intervals, the size of the samples was measured and compared with initial volumes. Three replicates were conducted for each TTH sample. In a control experiment, TTH samples were frozen at −20 °C and subsequently dried by lyophilization.

**Dissolution of TTHs with triggers**. The dissolution of TTHs was studied by using EDTA and GSH as triggers. Typically, the TTH were cut into 1 cm³-sized cubes and submerged in 10 ml PBS (pH 7.4) containing EDTA and GSH with a range of concentrations from 20 to 80 mM in a 20 ml VWR glass vial. Three replicates for each time point and condition were incubated at 37 °C on a shaker plate at 250 r.p.m. At each time point, the TTH cubes were subjected to compression measurement. The TTH cubes incubated in either 20 mM of EDTA or GSH were used as controls. To demonstrate the complete dissolution of TTHs into free polymer chains, the triggered solutions were filtered by a 0.2 μm filter and subsequently injected into GPC. For cytotoxicity assay of the dissociated polymers, the triggered solutions were transferred to dialysis tubes (MWCO, 10 kDa), then dialyzed against pure water for 3 days to remove EDTA and GSH, and finally dried by lyophilization.

**Permeability measurement**. The measurement of permeability of TTHs was carried out on a Franz diffusion cell using a TTH membrane (thickness, 3 mm). 2 ml PBS (pH 7.4) containing 1 mg ml⁻¹ DMSO, rifampicin, or insulin was added into the donor compartment of the cell and 12 ml fresh PBS was placed in the acceptor compartment of the cell. At each time point, 0.4 ml was sampled from the acceptor compartment and 0.4 ml fresh PBS was supplemented through the sampling port of the cell. The concentration of DMSO and rifampicin of the samples was recorded on a Perkin-Elmer Lambda ultraviolet-visible spectrometer, and the UV absorbance calibration curve of DMSO in a range from 6.25 to 100 μg ml⁻¹ or rifampicin in a range from 1.56 to 100 μg ml⁻¹ with a correlation coefficient >99.9% was used to determine the concentration. The

content of Alexa Fluor 488-labeled insulin was measured on an Infinite M200Pro (Tecan) reader (excitation, 490 nm; emission, 540 nm).

**In vitro drug release**. Individual 50 mg TTH cubes with lumefantrine content of 1, 5, or 10 wt% were used for long-term release studies. Typically, the TTH cubes were submerged in 2 ml SGF in a 15 ml VWR centrifuge tube and then incubated at 37 °C on a shaker plate at 250 r.p.m. At each time point, the release medium was replaced by 2 ml fresh SGF and then frozen at −80 °C until analysis. The release study was carried out for up to 12 days and the total drug release was measured by HPLC using a linear standard curve of lumefantrine with a range of concentration from 0.005 to 50 μg ml⁻¹ and a correlation coefficient >99.9%.

**Cytotoxicity assay**. The cytotoxicity assay of the dissociated polymers was conducted by adding the polymers directly into the culture medium with a range of concentrations from 0.02 to 5 mg ml⁻¹. For the TTH, the gel was incubated in the culture medium with a range of dosage from 0.2 to 50 mg ml⁻¹ at 37 °C for 24 h. The obtained medium was then tested for its toxicity toward cells. Cell lines were purchased from ATCC and Public Health England for these experiments. To avoid cross contamination, expanded cells were stored in individual containers. Regular mycoplasma evaluations were performed of the cell culture environment and the cell lines to ensure the absence of mycoplasma contamination. Cytotoxicity was tested on HeLa, C2BBe1 (ATCC), and HT29-MTX-E12 cells (Public Health England) by seeding them each in a 96-well plate at a density of 10,000 cells per well as described previously[13] with minor modifications. HeLa and HT29-MTX-E12 cells were cultured in 100 μl Dulbecco's modified Eagle medium (DMEM) (Life Technologies) containing 1% non-essential amino acids (Life Technologies), 10% fetal bovine serum (Life Technologies), and 1% penicillin-streptomycin solution (Life Technologies) per well. C2BBe1 cells were cultured in the same medium, but the 1% non-essential amino acids were replaced with 1% human insulin-transferrin-selenium (Life Technologies). Cells were kept in culture for 3 days before replacing the medium with 100 μl of the pre-prepared solutions as described above. After 24 h, these solutions were replaced with 100 μl untreated media and cytotoxicity was quantified by adding 10 μl alamarBlue reagent (Life Technologies) to each well. The contents were mixed and then allowed to incubate at 37 °C for 1 h. Fluorescence was recorded on an Infinite® M200Pro (Tecan) with excitation at 560 nm and emission at 590 nm. A positive control was provided by treating cells with 70% ethanol for 1 hour prior to the cytotoxicity assay experiment. Cells that were not subjected to any polymer-treated media provided a negative control. Cell viability was calculated by the following equation: cell viability (%) = 100 × [Absorbance(sample)−Absorbance(positive control)]/[Absorbance(negative control)−Absorbance(positive control)].

**Stem cell culture**. Cell culture media (Advanced DMEM/F12 with N2, B27, and N-acetylcysteine) containing growth factors (EGF, Noggin, and R-Spondin 1) and small molecules (CHIR99021 and VPA) were used for stem cell culture. All animal experiments were performed in accordance with protocols approved by the Committee on Animal Care at MIT. Single mouse Lgr5-GFP ISCs were isolated from Lgr5-EGFP-IRES-CreERT2 mice (Jackson Labs) as described previously[50]. The isolated single Lgr5-GFP stem cells were cultured in Matrigel for 2 days to form stem cell colonies before use. To evaluate the cytobiocompatibility of TTHs against stem cells, the TTH dishes (thickness: 1 mm, diameter: 8 mm) were cultured directly with Lgr5-GFP stem cell colonies in a 24-well plate for 5 days. To test the ability of TTHs serving as a substrate for organoid culture, Lgr5-GFP stem cell colonies were mixed with Matrigel and then placed on the lyophilized TTH dishes in a 24-well plate. The plate was either placed in a 37 °C incubator directly (for cell culture on the TTHs), or further incubated for 30 min on ice and subsequently centrifuged at 300 × g for 2 min before placed in a 37 °C incubator (for cell culture within the TTHs). Cells were cultured in stem cell culture media for 3 days before switching to organoid culture media (by removing CHIR99021 and VPA to permit spontaneous differentiation of the stem cells) and further cultured for another 4 days.

**In vivo studies**. All animal experiments were performed in accordance with protocols approved by the Committee on Animal Care at MIT and as previously described[13, 45] with minor modifications. A large animal model, 45–55 kg Yorkshire pigs, was chosen as its gastric anatomy similar to humans and is widely used in evaluating devices in the GI space. Pigs were sedated with Telazol (tiletamine/zolazepam) 5 mg kg⁻¹, xylazine 2 mg kg⁻¹, and atropine 0.05 mg kg⁻¹, and/or isoflurane (1-3% inhaled), and an endoscopic overtube (US Endoscopy) was placed in the esophagus under endoscopic visual guidance during esophageal intubation. To evaluate the TTH for its ability to be triggered to dissolve into solution with biocompatible agents, the retrievable TTH strips and methyl blue-labeled large TTH sheets were administered via the overtube into the stomach. PBS (0.5 l) containing EDTA (40 mM), GSH (20 mM), and NaHCO₃ (60 mM) was administered via the overtube after the gastric placement of the TTH samples. Intra-gastric endoscopy videography was used for image capture of the dissolution of the TTH sheets. The TTH strips were retrieved endoscopically after 1 h in the gastric cavity. Pigs that were not administered EDTA/GSH and therefore the strips and sheets were only exposed to gastric fluid were used as control experiments. To assess

TTHs for the ability to achieve gastric retention, radiopaque barium sulfate-labeled capsule-like TTHs were administered via the overtube into the gastric cavity (one TTH device per pig). Radiographs were performed every 48–72 h to monitor the integrity and transit of the devices as well as any radiographic evidence of bowel obstruction or perforation. In vivo drug release experiments were performed with dosage forms (one drug-loaded TTH device in their hydrated states containing 960 mg of lumefantrine per pig) in the same dimensions and shape to the barium sulfate-loaded TTH device. Blood samples were obtained via cannulation of an external mammary vein on the ventral surface of the pig at indicated time points, most often time 0 min (prior to administration of the dosage form), 5 min, 15 min, 30 min, 2 h, 6 h, and then daily for a minimum of 5 days and then three times per week. During the evaluation of the TTH systems for gastric residence and drug delivery, the animals were monitored twice daily for any signs of abnormal feeding and stooling patterns. In addition, the animals were monitored clinically for any evidence of GI obstruction as well as radiographically every 48–72 h for evidence of obstruction and/or perforation.

**Modeling pharmacokinetic data**. To determine the elimination rate constant and half-life of the free lumefantrine, a one-compartment oral absorption model was fit to the pharmacokinetic data. The model is shown in Supplementary Fig. 19a. Using this model, an equation was derived to describe the plasma concentration time profile.

$$C_p = \frac{\text{Dose}\, k_a}{V(k_a - k_e)}\left[e^{-k_e t} - e^{-k_a t}\right] \tag{1}$$

To describe the pharmacokinetic profile obtained from the lumefantrine-loaded TTH device, a pharmacokinetic model described in Supplementary Fig. 19b was used. An equation was obtained to describe the plasma concentration.

$$C_p = \frac{\text{Dose}\, k_{rel} k_a}{V}\left[\frac{e^{-k_{rel} t}}{(k_a - k_{rel})(k_e - k_{rel})} - \left(\frac{1}{k_a - k_e}\right)\left(\frac{e^{-k_e t}}{k_{rel} - k_e} - \frac{e^{-k_a t}}{k_{rel} - k_a}\right)\right] \tag{2}$$

**Data availability**. The data supporting the findings of this study are available within the article and its Supplementary Information files and from the corresponding authors on reasonable request.

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

## Acknowledgements

This work was funded in part by the Bill and Melinda Gates Foundation Grants OPP1096734 and OPP1139927 and the NIH Grant EB000244. The paper was partly sponsored by the Alexander von Humboldt Foundation under the auspices of the Max Planck Research Award to R.L. funded by the Federal Ministry of Education and Research.

## Author contributions

J.L. designed the material and experiments. R.L. and G.T. supervised the research. J.L. prepared the material and the device. J.L., Y.P., S.Z., C.C., X.Y., L.B., J.L., Y.-A.L.L., H.M., S.S., A.R.K., T.E., J.R.,R.L., and G.T. characterized the material and analyzed the data. J.L., R.L., and G.T. wrote the paper. All authors discussed the progress of research and reviewed the manuscript.

## Additional information

**Competing interests:** J.L., R.L., and G.T. are co-inventors on a provisional patent application number 62/419650 describing the TTHs, which was filed on 9 November 2016. The remaining authors declare no competing financial interests.

