## [Peer Review File · Nature Communications]

Reviewer #1 (Remarks to the Author):

This manuscript describes a novel hydrogel material of triggerable tough hydrogels (TTHs) with the ability to reside in the gastric cavity for diagnosis and treatment interventions in humans, for which the materials conventionally used for their fabrications have been largely limited to thermoplastic and thermosets. Authors claimed that TTHs contain up to ~90% water enabling significant shrinkage in a form factor compatible with ingestion and subsequently swell to change shape to enable gastric residence and prevent passage through the pylorus. These gels also demonstrate stability in normal gastric environments and furthermore can be triggered to dissolve rapidly with biocompatible agents, enabling the development of gastric devices with precisely controlled residence times. A proof of concept of designing gastric resident drug delivery system based on TTHs has showed drug released from these materials in a sustained manner in a large animal model. However, gastric resident drug delivery systems developed in pharmaceutical field mostly used hydrogel materials, such as high viscosity grade of HPMC and high molecular weight polyethylene oxide (PEO). Those hydrogel materials can be easily produced as a tablet form with a high drug loading. Tablets so produced are administered orally without difficulty and is able to swell to a size greater than the diameter of pylorus after ingestion to extend its residence time in stomach. Swelling hydrogels also possess a gel strength that highly resistant to the mechanical force exerted by stomach contraction and is able to gradually dissolved without any need to trigger its breakage. Therefore, the novelty of using TTHs as gastric resident drug delivery systems would be inadequately. To include another example of applying TTHs to treatment intervention or tissue engineering might increase its novelty. Overall, before considering to accept this manuscript for publication, the following points needed to be addressed:

1. Line 95-100: How could alginate not be crosslinked by Ca^{2+} to form one of the two networks when all ingredients including calcium sulfate were dissolved in deionized water before subjecting the reaction mixture to 50 °C for 1 h?
2. Line 118-121: Although authors claimed that the TTH swelled progressively and the plateau of volume variation of 2.70.15 was reached at 6 days, the most important characteristics for those gastric resident dosage forms based on swelling mechanism is to swell to the extent that larger than the diameter of pylorus within 15 min immediately after injection in order to prevent gastric resident dosage form from getting passed pylorus?
3. Line 137-138: Authors were suggested to indicate how soon can TTH recover to a diameter larger than that of pylorus instead of describing that TTH recovered to initial volume within 6 hrs after rehydration?
4. Line 169-170: What was the underlying mechanism responsible for TTH being not dissolved by incubation with EDTA or GSH alone?
5. Line 181-184: How did GSH affect the binding between EDTA and Ca^{2+} cation?

6. Line 208-210: It would be impractical if a trigger solution was needed to be administered to dissolve TTH in vivo?
7. Line 215-219: Why not orally administered dehydrated TTH containing barium sulfate to examine its rehydration, gastric retention, and integrity in vivo?
8. Line 236-238: What was the rationale of selecting lumefantrine as the model drug to be tested in TTHs gastric resident dosage form?
9. Line 240-241: Authors claimed that the degree of the drug loading was easily controlled by feed ratio of drug. However, there was a washing step after formation of TTHs. How significant of this washing step affecting the final drug loading in TTH? For those water soluble drugs, this washing step could result in a greater extent of drug loss expectedly?
10. Line 246-250: Authors claimed that it was due to hydrophobicity of TTH with increasing drug loading leading to a slower release rate? However, It might be potentially due to the retardation of TTH swelling with increasing hydrophobicity of increasing drug loading? It was suggested to present the swelling profile for TTH containing different weight % of drug for comparison?
11. Line 258-259: Was it a dehydrated form of TTH device containing 960 mg of lumefantrine administered to each pig for PK studied?

Reviewer #2 (Remarks to the Author):

This is a very interesting manuscript on a novel type of advanced drug delivery systems, allowing for prolonged residence times in the stomach, combined with time-controlled drug release kinetics. This type of dosage forms can be beneficial for the treatment of various diseases.

The paper is very well written and easy to follow. The results are very interesting and the conclusions justified by the presented data.

I have only the following minor comments:

1) In vitro drug release kinetics:

Since lumefantrine is a hydrophobic drug and the volume of the release medium was limited (2 mL, replaced at each sampling time point), might saturation effects play a role for the observed in vitro release kinetics?

If this is the case, the observed decrease in the relative drug release rate in vitro with increasing drug loading (Figure S15) might at least partially be attributable to limited drug saturation of the release medium. It might, hence, eventually be interesting to measure drug release in vitro also upon exposure to larger volumes of bulk fluid.

2) In vivo study:

It would be interesting to indicate the drug loading (in percent) of the system that was tested in vivo, and to briefly comment on the release rates observed “in vitro” versus “in vivo”.

3) Pharmacokinetic analysis:

a) It seems that first order kinetics were used to describe drug transport from one compartment to the other (Figure S17, at least for the absorption and elimination kinetics, according to the equation indicated in line 504). It would be good to mention this. Also, it would be interesting to indicate all the constants, which have been determined (including the values for k_{rel} and k_a), and to add the units of the elimination rates constants in the legend of Figure S17.

b) It might be worth trying to fit an equation describing first order release kinetics to the experimentally measured in vitro drug release kinetics shown in Figure S15. If good agreement is obtained, these in vitro release rate constants could be compared to the release rate constant determined in vivo (k_{rel} , Figure S17).

c) Theoretically, the elimination rate constant of the drug (k_e) should be the same upon administration of “free” drug and administration of the drug-loaded gastroretentive delivery system (since this value should be drug-specific). It seems that the k_e -value was affected by the prolonged release of the drug from the gastroretentive delivery system. It would be good to briefly mention/comment on this, and to maybe call k_e “apparent elimination rate constant” in this case.

Reviewer #3 (Remarks to the Author):

Gastro-retentive dosage forms for modulated release of drug is need of hours. The present topic is found to be interesting and selected methodology is appropriate to address the issue of proposed objectives. Authors have puts rational and intensive efforts to justify the selection of hybrid polymeric film for gastroretentive drug delivery systems using suitable in-vitro-in-vivo experimental approaches. However there are few comments required to be addressed:

- Considering the extended gastroretentive behavior of the developed model, 24 hrs cytotoxicity study might not be enough to account for the toxicity of developed systems.
- Developed polymeric systems displayed extended gastric retention. Therefore, it is necessary to evaluate the vital gastric function, including gastric secretion, digestive function, and gastric emptying rate.
- The author used radical polymerization to prepare a co-polymer. However, for theoretical and practical reasons, it is of interest to discuss the reaction conditions to achieve stereoselectivity of polymerization, which is not discussed. The method should also include an appropriate analytical evaluation to interpret the chemical aspects of crosslinking.
- In this study, lumefantrine was selected as a model drug, which is practically insoluble in water. Results indicated that physical and pharmaceutical properties of the prepared formulation were not affected by increasing drug loading from 1 to 10%. This is difficult to understand and requires compelling evidence for a true understanding of the mechanism?
- It is difficult to understand the SD values (33 MPa and 17 MPa) reported for hydrogels composed of polyacrylamide or alginate, respectively.

Point-by-point response / NCOMMS-16-30409

Reviewer #1:

This manuscript describes a novel hydrogel material of triggerable tough hydrogels (TTHs) with the ability to reside in the gastric cavity for diagnosis and treatment interventions in humans, for which the materials conventionally used for their fabrications have been largely limited to thermoplastic and thermosets. Authors claimed that TTHs contain up to ~90% water enabling significant shrinkage in a form factor compatible with ingestion and subsequently swell to change shape to enable gastric residence and prevent passage through the pylorus. These gels also demonstrate stability in normal gastric environments and furthermore can be triggered to dissolve rapidly with biocompatible agents, enabling the development of gastric devices with precisely controlled residence times. A proof of concept of designing gastric resident drug delivery system based on TTHs has showed drug released from these materials in a sustained manner in a large animal model.

We thank the reviewer for their constructive comments and suggestions, which we have addressed below and have helped strengthen our manuscript.

However, gastric resident drug delivery systems developed in pharmaceutical field mostly used hydrogel materials, such as high viscosity grade of HPMC and high molecular weight polyethylene oxide (PEO). Those hydrogel materials can be easily produced as a tablet form with a high drug loading. Tablets so produced are administered orally without difficulty and is able to swell to a size greater than the diameter of pylorus after ingestion to extend its residence time in stomach. Swelling hydrogels also possess a gel strength that highly resistant to the mechanical force exerted by stomach contraction and is able to gradually dissolved without any need to trigger its breakage. Therefore, the novelty of using TTHs as gastric resident drug delivery systems would be inadequately.

To address these points we have expanded the Introduction recognizing and highlighting developments in the gastric resident dosage form field and placed our contributions in the context of those developments. We have noted that using high viscosity grade of cellulose polymers, polyacrylic acid, polyacrylates, polyacrylamides, and polyethers extended release of drugs, on the order 12-16 hours, has been previously achieved. These systems do not reside long term in the gastric cavity largely due to the weakness and brittleness of these hydrogel materials. In contrast, the TTH system achieved significantly prolonged gastric residence with an intact dosage form on the order of 7 to 9 days as demonstrated in a large animal model secondary to the stiffness and toughness of TTHs. As the reviewer noted the ability to withstand the environment and gastric forces are essential to achieving long-term residence.

For systems which transit out of the gastric cavity and/or out the entire gastrointestinal tract safety concerns may be reduced and therefore the requirement for triggering

dissolution may not be essential. One of the goals of the systems we have described is to provide a system that can reside within the body on the order of a week or longer. Given these requirements clinically the development of either gastric outlet obstruction and/or an allergic reaction would necessitate immediate intervention and therefore a system that can be triggered would have significant advantages over ones without the capacity for triggering. Specifically, a system that can be triggered to dissolve could be intervened upon with the ingestion of a solution and one that doesn't would require either endoscopic or surgical intervention.

With respect to the performance characteristics of TTHs to ensure their gastric retention we verified that the TTHs were compatible with ingestion and subsequent rapid expanding to a size greater than the diameter of the pylorus within the emptying time of human stomach.

Taken together, TTHs demonstrate an innovative hydrogel material with exceptional ability to withstand the environment and gastric forces as well as significantly improved safety profile for extended (well beyond the currently approved hydrogel systems) oral drug delivery. We believe this work represents a significant advance in research area of drug delivery, biomaterials, and translational medicine.

To include another example of applying TTHs to treatment intervention or tissue engineering might increase its novelty. Overall, before considering to accept this manuscript for publication, the following points needed to be addressed.

We thank the reviewer for their suggestion to expand the application of the TTH system. We have added significant data supporting the potential application of TTHs for tissue engineering serving as a substrate for organoid culture. Specifically, we demonstrate excellent cytocompatibility of the TTHs with mouse Lgr5⁺ intestinal stem cells (ISCs). Furthermore, we demonstrate that Lgr5⁺ stem cells can be cultured on and within TTHs and they retain their ability of multilineage differentiation to form organoids. We have added this data to Figure S8 in the supplementary information.

1. Line 95-100: How could alginate not be crosslinked by Ca²⁺ to form one of the two networks when all ingredients including calcium sulfate were dissolved in deionized water before subjecting the reaction mixture to 50 °C for 1 h?

Calcium sulfate was added as a suspension into the reaction mixture because of its limited water solubility caused by its low dissociation constant. Although the association of Ca²⁺ with the carboxyl groups in alginate could accelerate the dissolution of calcium sulfate, the complete dissolution took place overnight. Thus the reaction mixture was presented as a free solution before subjecting it to polymerization even after all the ingredients were added. This has been clarified in the Experimental Section.

2. Line 118-121: Although authors claimed that the TTH swelled progressively and the plateau of volume variation of 2.70.15 was reached at 6 days, the most important characteristics for those gastric resident dosage forms based on swelling mechanism

is to swell to the extent that larger than the diameter of pylorus within 15 min immediately after injection in order to prevent gastric resident dosage form from getting passed pylorus?

We thank the reviewer for highlighting this point, which we have now expanded upon with further experimentation and clarified in the text. Briefly, in Figure 2f, a completely dehydrated TTH with similar dimensions to a standard triple zero capsule is shown to swell in simulated gastric fluid to a size greater than the diameter of the pylorus 12.8 mm within 70 minutes, which is within the 50 percentile for gastric emptying in humans. Additionally, we provide a strategy and experimental evidence supporting the capacity of a TTH that can swell to a size larger than the diameter of pylorus within 15 minutes in a neutral pH approximating the fed state or patients taking antacids or that can be achieved by co-administration with antacids. The enhanced swelling is attributed to the higher solubility of alginate in neutral pH than in an acidic environment. An alternative strategy is also demonstrated where a TTH-based encapsulation system is applied to encase CaCO_3 in an initial form factor of a standard triple zero capsule that can swell to 27 mm within 30 minutes in simulated gastric fluid (Figure S6, top). Similar strategies can be applied for enabling flotation as shown in Figure S6 (bottom).

Given the highly stretchable and tough characteristics, various dosage forms with properties enabling extended gastric residence can be developed by using TTHs.

3. Line 137-138: Authors were suggested to indicate how soon can TTH recover to a diameter larger than that of pylorus instead of describing that TTH recovered to initial volume within 6 hrs after rehydration?

We have addressed this under point 2 above.

4. Line 169-170: What was the underlying mechanism responsible for TTH being not dissolved by incubation with EDTA or GSH alone?

We have clarified this point in the text and in the supplementary information sections.

Briefly, the TTH is a double network hydrogel consisting of alginate and polyacrylamide networks that are intertwined, and separately crosslinked by stimuli-responsive Ca^{2+} ionic and disulfide bonds. To dissolve the TTH, both the alginate and polyacrylamide networks must be de-crosslinked simultaneously.

As shown in Figure S9, dissolution studies support that the TTH could not be dissolved by incubation with EDTA or GSH alone even when incubation times were increased to 24 h, indicating the other network was still crosslinked forming a single network hydrogel. This data demonstrate that de-crosslinking of both alginate and polyacrylamide networks are essential to dissolve the TTH.

5. Line 181-184: How did GSH affect the binding between EDTA and Ca^{2+} cation?

The binding between EDTA and Ca^{2+} is ascribed to the formation of ionic bond between the Ca^{2+} and the carboxyl groups in EDTA. We speculated that the carboxyl group at the C-terminus of GSH could disturb the formation of ionic bond between the Ca^{2+} and the carboxyl groups in EDTA when excessive GSH was present.

6. Line 208-210: It would be impractical if a trigger solution was needed to be administered to dissolve TTH in vivo?

We thank the reviewer for bringing this to our attention and have clarified the clinical value of triggering dissolution of a gastric resident system in our manuscript.

Long-term (>24 hr) gastric resident systems present risks to patients including: gastrointestinal mechanical obstruction and the inability to discontinue a drug in the event of developing an allergic reaction through non-invasiveness means.

The ability to trigger the dissolution of such systems is therefore essential for safe clinical implementation. The need for triggering is further amplified in resource constrained settings where healthcare interventions like endoscopy and surgery may be largely limited and where the inability to remove such systems could manifest in significant morbidity and mortality.

7. Line 215-219: Why not orally administered dehydrated TTH containing barium sulfate to examine its rehydration, gastric retention, and integrity in vivo?

We thank the reviewer for highlighting this point. We have clarified in the text the limitations associated with barium-containing gels. Specifically, the significant load of barium required for radiographic visualization (20 wt%) manifested in slower swelling characteristics than the non-barium containing gels (Figure S14). We therefore used barium-containing TTHs in their hydrated states which enabled the retention by virtue of the size of the gel administered and radiographic visualization by virtue of their barium content.

8. Line 236-238: What was the rationale of selecting lumefantrine as the model drug to be tested in TTHs gastric resident dosage form?

We have clarified this point in the text. Briefly, medication non-adherence is a major challenge for the treatment of malaria and having the capacity to delivery drugs in a single administration event has the potential to not only enhance cure rates in acute malaria but also decrease resistance rates.

9. Line 240-241: Authors claimed that the degree of the drug loading was easily controlled by feed ratio of drug. However, there was a washing step after formation of TTHs. How significant of this washing step affecting the final drug loading in TTH? For those water soluble drugs, this washing step could result in a greater extent of drug loss expectedly?

We agree with the reviewer and have provided further clarification and data in the manuscript to clarify these points.

The washing step did affect lumefantrine loading of TTHs. According to the release data in Figure S17 (top left), the drug loading decreased to 0.56%, 4.57%, and 9.72% for the TTHs with drug loading of 1%, 5%, and 10% respectively after 24 h incubation. However, no drug was lost during the preparation of rifampicin-loaded TTHs. As shown in Figure S17 (bottom), we first prepared the purified TTH, then lyophilized and subsequently rehydrated the TTH in the aqueous solution of rifampicin (a water soluble antibiotic).

10. Line 246-250: Authors claimed that it was due to hydrophobicity of TTH with increasing drug loading leading to a slower release rate? However, It might be potentially due to the retardation of TTH swelling with increasing hydrophobicity of increasing drug loading? It was suggested to present the swelling profile for TTH containing different weight % of drug for comparison?

We thank the reviewer for this helpful suggestion. Accordingly, we carried out the measurement of the swelling profile of TTH containing different weight% of lumefantrine in simulated gastric fluid for comparison. The slower release rate was confirmed with increasing drug loading supporting the retardation of the TTH swelling correlated with a rise in hydrophobicity (Figure S17, top right).

11. Line 258-259: Was it a dehydrated form of TTH device containing 960 mg of lumefantrine administered to each pig for PK studied?

We have clarified this in the Experimental Section that a hydrated form of TTH device containing 960 mg of lumefantrine was administered to each pig for PK studies.

Reviewer #2

This is a very interesting manuscript on a novel type of advanced drug delivery systems, allowing for prolonged residence times in the stomach, combined with time-controlled drug release kinetics. This type of dosage forms can be beneficial for the treatment of various diseases.

The paper is very well written and easy to follow. The results are very interesting and the conclusions justified by the presented data.

I have only the following minor comments.

We thank the reviewer for their positive review of our work and for providing helpful suggestions on how to improve the quality of our manuscript.

1) In vitro drug release kinetics:

Since lumefantrine is a hydrophobic drug and the volume of the release medium was limited (2 mL, replaced at each sampling time point), might saturation effects play a role for the observed in vitro release kinetics? If this is the case, the observed

decrease in the relative drug release rate in vitro with increasing drug loading (Figure S15) might at least partially be attributable to limited drug saturation of the release medium. It might, hence, eventually be interesting to measure drug release in vitro also upon exposure to larger volumes of bulk fluid.

We thank the reviewer for bringing this point to our attention. Release experiments were carried out under a predetermined sink conditions. HPLC measurements suggested that the concentration of lumefantrine in all release media was much lower than the water solubility of lumefantrine (2 $\mu\text{g/mL}$, Afr. J. Med. Med. Sci. 2013, 42, 209–214), indicating the release media containing lumefantrine was unsaturated. In addition, we supplemented the measurement of the swelling profile of TTH containing different weight% of lumefantrine in simulated gastric fluid for comparison and found that the slower release rate was mainly ascribed to the retardation of the TTH swelling with increasing hydrophobicity associated with increasing drug loading (Figure S17, top right).

2) In vivo study:

It would be interesting to indicate the drug loading (in percent) of the system that was tested in vivo, and to briefly comment on the release rates observed “in vitro” versus “in vivo”.

We have clarified this in the text. The drug loading of the lumefantrine-loaded TTHs tested in vivo was 4.1 wt%. With respect to the TTHs with similar drug loading, the release rate estimated through in vitro study was lower than that obtained from the in vivo pharmacokinetic study. This may be due to food effects as well as those from gastric secretions which may enhance the rate of release of the hydrophobic drug from the TTH.

3) Pharmacokinetic analysis:

a) It seems that first order kinetics were used to describe drug transport from one compartment to the other (Figure S17, at least for the absorption and elimination kinetics, according to the equation indicated in line 504). It would be good to mention this. Also, it would be interesting to indicate all the constants, which have been determined (including the values for k_{rel} and k_a), and to add the units of the elimination rates constants in the legend of Figure S17.

We thank the reviewer for their suggestions. The values of the various constants estimated by the pharmacokinetic model have now been added to the main manuscript and Figure S19.

b) It might be worth trying to fit an equation describing first order release kinetics to the experimentally measured in vitro drug release kinetics shown in Figure S15. If good agreement is obtained, these in vitro release rate constants could be compared to the release rate constant determined in vivo (k_{rel} , Figure S17).

We have incorporated this suggestion and an equation describing first order release kinetics was fit to the in vitro release data. The rate constant for drug release estimated through this study was lower than that obtained from the pharmacokinetic study. This

may be because of food effects and effects from the gastric secretions which may enhance the rate of release of the hydrophobic drug from the gel.

c) Theoretically, the elimination rate constant of the drug (k_e) should be the same upon administration of “free” drug and administration of the drug-loaded gastroretentive delivery system (since this value should be drug-specific). It seems that the k_e -value was affected by the prolonged release of the drug from the gastroretentive delivery system. It would be good to briefly mention/comment on this, and to maybe call k_e “apparent elimination rate constant” in this case.

We agree with the reviewer suggestion and have now included the units for the rate constant and also use the term “apparent elimination rate constant”.

Reviewer #3

Gastro-retentive dosage forms for modulated release of drug is need of hours. The present topic is found to be interesting and selected methodology is appropriate to address the issue of proposed objectives. Authors have puts rational and intensive efforts to justify the selection of hybrid polymeric film for gastroretentive drug delivery systems using suitable in-vitro-in-vivo experimental approaches. However there are few comments required to be addressed:

We thank the reviewer for their suggestions, which we have addressed below.

Considering the extended gastroretentive behavior of the developed model, 24 hrs cytotoxicity study might not enough to account the toxicity of developed systems.

We thank the reviewer for this point. Given cell passage is needed every two days, the cytotoxicity assay is not suitable for long-term studies by using these cells. To address this point, we conducted extended cytotoxicity analysis over 5 days by culturing the TTHs with stem cells. As shown in Figure S8, the TTHs showed excellent cytocompatibility with intestinal stem cells (ISCs) over the course of 5 days.

Developed polymeric systems displayed and extended gastric retention. Therefore it is necessary to adjudge the vital gastric function including gastric secretion, digestive function and gastric emptying rate.

We thank the reviewer for bringing this to our attention and have clarified in the text the ability of the TTH dosage forms to remain stable in the gastric cavity. Additionally we have clarified the level of evaluation the large animals undergo which involves monitoring of feeding, stooling patterns, weight and clinical evidence of gastrointestinal obstruction in the Experimental Section. Additionally, animals are monitored radiographically for any evidence of gastrointestinal perforation and obstruction.

Author used radical polymerization technique to prepare co-polymer. However, in theoretical and practical reason it is of interest to discuss the reaction condition to

achieve stereoselectivity of polymerization, which is not discussed. In addition method should also include an appropriate analytical evaluation to interpret the chemical aspect of crosslinking.

The monomers of acrylamide and *N,N'*-bis(acryloyl)cystamine were randomly copolymerized by radical polymerization. As far as we know, the polymer chain synthesized by this approach has no stereoselectivity.

Regarding the chemical aspects of the crosslinking, we think available methods for efficient characterization of crosslinking are quite limited due to the low concentration of the crosslinker monomer (0.038 wt%). However, we demonstrated the successful introduction of disulfide crosslinks into the gel. As shown in Figure S9, the TTHs were sensitive to reductive GSH indicating the existence of disulfide bonds in the polyacrylamide network. In addition, only a viscous solution other than solid gel was obtained when the radical polymerization was carried out without adding *N,N'*-bis(acryloyl)cystamine in a control experiment.

In this study lumefantrine was selected as model drug, which is practically insoluble in water. Results indicated that physical and pharmaceutical properties of prepared formulation were not affected by increasing drug loading from 1 to 10%. Which is difficult to understand and required to compiling evidence for true understanding of the mechanism?

We have clarified this further in the text. Our results showed that the physical and pharmaceutical properties of the lumefantrine-loaded TTHs were affected by increasing drug loading from 1 to 10 wt%. First, the maximum compressive stress of the gel increased from 3.91 ± 0.31 to 5.43 ± 0.61 MPa with the increase of drug loading from 1 to 10 wt%, whereas the fracture strain decreased from 14.7 ± 1.3 to 11.9 ± 1.5 (Figure S16). Second, in vitro cumulative release of lumefantrine after 12 days incubation in simulated gastric fluid increased from $8.3 \pm 0.17\%$ to $61 \pm 3.7\%$ with the decrease of drug loading from 10 to 1 wt% (Figure S17, top left). Lastly, the swelling of TTH containing different lumefantrine loading in simulated gastric fluid retarded with increasing hydrophobicity of increasing drug loading from 1 to 10 wt% (Figure S17, top right).

It difficult to understand the SD value (33 MPa and 17 MPa) reported for hydrogels composed of polyacrylamide or alginate respectively.

We are grateful to the reviewer for drawing our attention to these typos. We have corrected the “ 0.275 ± 33 MPa” and “ 0.121 ± 17 MPa” to “ 0.275 ± 0.033 MPa” and “ 0.121 ± 0.017 MPa” respectively in our revised manuscript.

Reviewer #1:

Remarks to the Author:

The points raised in the previous round of review have been satisfactorily addressed by the corresponding author.

Reviewer #2:

Remarks to the Author:

All comments have been addressed in an appropriate manner.

Reviewer #3:

Remarks to the Author:

Authors have made all the correction in revised manuscript. Now manuscript is acceptable for publication.

NCOMMS-16-30409A – Triggerable tough hydrogels for gastric resident dosage forms

Reviewer #1 (Remarks to the Author):

The points raised in the previous round of review have been satisfactorily addressed by the corresponding author.

Reviewer #2 (Remarks to the Author):

All comments have been addressed in an appropriate manner.

Reviewer #3 (Remarks to the Author):

Authors have made all the correction in revised manuscript. Now manuscript is acceptable for publication.

Response: We thank the reviewers for their helpful and constructive comments that have helped improve our manuscript.